# Effects of Mellowing Practice on the Strength and Swelling Properties of Road Construction Materials: Case of Sulphate-Bearing Clay Soils Stabilised with Lime-Silica Fume Blended Binder

**DOI:** 10.3390/ma16062187

**Published:** 2023-03-09

**Authors:** Qusai Al-Waked, John M. Kinuthia, Blessing O. Adeleke, Jonathan Oti, Ahmed Khalifa

**Affiliations:** 1Faculty of Computing, Engineering and Science, University of South Wales, Pontypridd CF37 1DL, UKjohn.kinuthia@southwales.ac.uk (J.M.K.);; 2Elkem ASA, Kristiansand, 8040 Vaagsbygd, Norway

**Keywords:** ettringite, mellowing, calcium-based stabilizer, silica fume, compressive strength, linear expansion, swelling

## Abstract

The main thrust of this research was to establish any benefits of mellowing, and the optimal moisture content (OMC) for compacting mellowed sulphate-bearing clay soil undergoing the stabilization process. Two three-day mellowing regimes were carried out, prior to final compaction, at different initial moisture contents of 30% or 40% OMC. The unmellowed specimens were compacted immediately after mixing with the blended stabilizers. A blend of quicklime (L) with a sustainable by-product, silica fume (SF), at a 1:1 ratio (2%L–2%SF) was used. Linear expansion and unconfined compressive strength (UCS) tests were carried out to evaluate the benefits of mellowing. The test results suggested that the mellowed test specimens achieved better UCS and swelling properties compared to the unmellowed specimens. Mellowing at 1.2 OMC produced better strength performance than at 1.4 OMC, whereas mellowed specimens at 1.4 OMC showed better resistance to linear expansion compared to 1.2 OMC. The research findings suggested that optimal performance was achieved by mellowing at the higher initial moisture condition of 40% OMC and compacting the materials at the lower moisture condition of 1.2 OMC.

## 1. Introduction

The quality of sub-grade soils in road construction is vital to roads’ durability [1]. Sub-grade soils with low load-bearing capacity soil, and/or high expansion properties are considered unsuitable for road construction [1]. Swelling of some clayey soils due to changes in water content may lead to detrimental movement of the foundation structures, ultimately resulting in an excessive settlement, shear failure, and heaving [1]. Expansive soils are reported to be of particular concern for road pavement. Sulphate-bearing soils experience loss in strength and high changes in volume, leading to significant heave in the stabilized earthwork [2].

Extensive research has been carried out on the effects of sulphate as either naturally present in the ground soil or artificially dosed in term of soil stabilization with lime and/or cement [3,4,5,6,7,8]. This heave or expansion in lime-stabilized clay soil in the presence of sulphate might be partially attributed to the growth of ettringite crystals formed on the clay particle surfaces [3]. Soil stabilization for road construction using calcium-based stabilizers such as lime has been widely used to mitigate the shrink-swell behavior of expansive clay [9]. However, lime production is of great concern due to the higher energy consumption (4000 MJ/ton) and the high carbon dioxide footprint (800 kg CO_2_eq/ ton) associated with lime production [10]. In addition, lime has been reported to have limited efficacy for soil stabilization in the presence of sulphate, as it may lead to serious cases of swelling due to the formation and growth of ettringite [11]. In this context and to improve the sustainability credentials of the construction industry, industrial and agricultural waste streams have been studied for soil stabilization. Pulverized fuel ash (PFA), ground granulated blast-furnace slag (GGBS), and calcined kaolin clay, metakaolin (MK), have been the most commonly used reactive pozzolanic materials.

One of the earliest reported studies on the use of GGBS in road construction to suppress sulphate-induced swelling was carried out by Wild et al. [3]. However, in Europe, sustainable future supplies of both PFA and GGBS have become questionable [12]. Thus, there is a need to find alternative methods and materials/binders to PFA and GGBS with an overall aim to stabilize both sulphate-bearing and non-sulphate-bearing clay soils. Extensive studies showed that using silica fume (SF) as a blend to lime and an alternative to GGBS could be a promising and encouraging binder to suppress swelling without compromising compressive strength. However, its high cost and availability have often dissuaded its usage [13,14,15].

Mellowing is mainly aimed at delaying the compaction of the wet lime-stabilized clay soil for a period of up to three days. This delay is believed to allow the clay soil in its moist condition to reach full modification from the effect of calcium ions present in the lime [16]. Currently, it is mandatory in the United Kingdom to ensure a 24–72-h period for mellowing of clay soils in road construction while carrying out stabilization using quicklime. This is in line with the Department of Transport Specification for Highway Works (1991 and 1993 amendments) [17].

Normally, soil samples are compacted wet at slightly higher OMC than the optimum moisture conditions of the clay soil to accommodate the cementitious binders used in the mixing process. This is carried out to cater for any moisture loss and to lower potential possible future swelling [18]. Therefore, soils are typically compacted at 1.2 OMC, in practice. In addition, the inclusion of cementitious binders usually introduces additional water demand, and thus water demand would be exacerbated if the mellowing process was adopted. Furthermore, mellowing would normally introduce further moisture losses leading to more complications in the compaction process [16]. During the mellowing process, only a partial amount of the total moisture content is needed in the initial stage of mixing the soil-binder-water mixture. This initial amount of water has never been effectively investigated or established. In practice, the initial amount of moisture content is typically used at approximately 30–40% of the total water needed, with the remainder of 60–70% being added after the mellowing period is ended (usually three days) during the final mixing and compaction [19].

The literature showed some contradiction in terms of the benefits associated with adopting mellowing during soil stabilization. Ali and Mohamed [20] examined the effects of using mellowing procedure (delay in compaction) for a period of 3, 6, 12, 24, and 48 h on a very high plastic clay with a plastic limit of 287%. They reported a reduction in UCS and an increase in swelling of the mellowed specimens. Similarly, Rahmat and Kinuthia [1] reported that the linear expansion for the mellowed specimens initially decreased with the increase in the lime content from 2% to 4%, but when the lime content increased to 6% the linear expansion increased. The results also indicated that the overall strength performance of the unmellowed, stabilized materials is superior to that of the mellowed specimens. On the contrary, Harris et al. [19] argued that employing a three-day mellowing period is an effective method that can lead to an acceptable swelling for sulphate-bearing clay soils consisting of sulphate content below 7000 ppm. Similar conclusions were found by Puppala et al. [21], who reported a reduction in swelling of mellowed, lime-stabilized, natural expansive soil.

To this end, the main thrust of this study was to investigate and establish the benefits of adopting mellowing, as well as the optimal moisture condition for mellowing kaolinite clay artificially dosed with 6 wt.% gypsum and stabilized using a 2%lime-2%SF blended binder. In addition, the effects of two different initial moisture content during mellowing were investigated for a better understanding of the effects of mellowing of sulphate-clay soil during the modification process. It is anticipated that the findings of this research, especially in the UK, should be of high interest to engineering geologists, practicing engineers, and researchers worldwide.

## 2. Experimental Methodology

### 2.1. Materials

For the fabrication of the test specimens, the following raw materials were used, Kaolinite-rich soil (K), gypsum (G), quicklime (L), and silica fume (SF). The kaolin clay soil was supplied in the form of a white fine powder, having a liquid limit of 56.7%, a plastic limit of 33.3%, and a plasticity index of 23.4%. It was supplied by Potterycrafts ltd, Stoke-on-Trent, UK. Its commercial trade name is China clay standard porcelain powder. The gypsum was supplied by Fisher Scientific Ltd., Loughborough, Leicestershire, Leicester, UK, in the form of a white fine powder with a purity of ≥90%. The quicklime used was provided by Tarmac Cement and Lime Ltd., Buxton Lime and Powders, Derbyshire, Debry, UK, in the form of an off-white fine powder. SF was supplied in the form of fine light grey amorphous powder, having a silicon dioxide content of 98.4%. SF was manufactured by Elkem in Norway, and was supplied by Tarmac Cement and Lime Company, Buxton Lime and Powders, Derbyshire, Derby, UK. Table 1 presents the oxide compositions of the raw materials.

The current research utilized the X-ray fluorescence (XRF) spectrometer to identify the chemical element composition for the materials in Table 1 (Kaolinite clay, Lime, and SF), in accordance with BS EN 15309:2007 [22] and BS ISO 18227:2014 [23]. The choice of using XRF spectrometry was based on its availability, non-consumptive technique, little or no pre-treatment, and ability to determine the total elemental concentrations in soils with a wide range of other matrix types (materials). Table 2 presents the physical properties of the raw materials.

The particle size distribution (PSD) of the materials in Table 2 was determined in accordance with BS EN ISO 17892-4:2016 [24] using a Marlvin Laser Diffraction Analyser (Mastersizer 2000), which uses infrared rays to determine the grain sizes of the soil particles. Each test was run 5 times and an average result of the particle size distribution was produced. The alkalinity values (pH) were obtained in accordance with BS EN ISO 787-9:2019 [25] using a portable pH/mV/0C meter with an accuracy of ±0.01, while others were sourced from existing literature with the same material source and production [26].

### 2.2. Mix Compositions and Specimens Preparation

Table 3 shows the mix composition designed for specimen preparation. Six different formulations were designed, comprising of kaolinite clay artificially dosed with 6 wt.% gypsum, and compacted at 1.2 or 1.4 OMC, as determined using the standard Proctor test. Stabilization was conducted with a blended binder comprising of 2 wt.% lime and 2 wt.% SF.

Two different mellowing procedures were adopted:Using water equivalent of 30 wt.% of the OMC, with the final compaction being carried out using the remainder of 70 wt.% of the OMC after mellowing.Using water equivalent of 40 wt.% of the OMC, with the final compaction being carried out using the remainder of 60 wt.% of the OMC after mellowing.

An artificial sulphate-clay soil was produced in the laboratory by blending kaolinite rich soil with gypsum at a dosage of 6% to simulate a high sulphate soil in practice. Gypsum was used based on the premise of being one of the most common sources of sulphate found in the natural sulphate soil. The stabilizer was compromised of a 1:1 lime-silica fume blended binder (2L-2SF). Proctor Compaction tests were carried out in accordance with BS 1377-4 [27] to determine the maximum dry density (MDD) and the OMC values of the kaolinite clay material. The MDD and OMC were established as 1.42 mg/m^3^ and 32%, respectively. However, a moisture content of 1.2% OMC of kaolinite rich soil was adopted for the unmellowed test specimens due to the impracticability of establishing the MDD and OMC values for each mix composition for both mellowed and unmellowed test specimens. In addition, it is worthwhile to note that normally a 3–5% additional water content above the optimum moisture content is recommended for mellowing [19]. Therefore, a 1.4% OMC of kaolinite rich soil was deemed appropriate to facilitate the compaction of the mellowed test specimens. This additional moisture content could be beneficial in the case of mellowing by delaying the setting time of the soil clay mixtures, thereby enhancing hydration during the mellowing period. In addition, the additional water could also accommodate any loss in moisture and accelerate the solubility of sulphates during the mellowing period.

The control mixes were prepared by weighing the required dry raw materials (soil and binders) and mixed in a mechanical mixer for 3 min. The required water was then added gradually and mixed for a further 3 min. The mixture was then poured into a steel mold of 100 mm in height by 50 mm in diameter. The mixture was then statically compressed using a hydraulic jack and the test specimen extruded using a plunger, and the test specimen was then wrapped in several runs of cling film, to conserve moisture. Finally, the test specimens were placed in a sealed air-tight container for moist curing at a temperature of 20 ± 2 °C until the testing date. The mellowed test specimens were prepared following the same procedure, the only difference being the use of additional water. Since two different initial moisture contents were adopted for the mellowed specimens, the mellowed specimens were initially mixed with 30% or 40% of the optimum moisture content. The mixture was then stored in a sealed container for 3 days. After 3 days of mellowing, the remainder of the water (70% or 60% OMC) was added to the mix and mixed thoroughly in the mechanical mixer prior to final compaction.

### 2.3. Experimental Testing Methods

#### 2.3.1. Unconfined Compressive Strength (UCS)

The UCS test was carried out using a 10 kN Hounsfield Testing Machine, in accordance with BS 1924-2:2018 [28], and BS EN ISO 17892-7:2018 [29] at a strain ratio of 2 mm per minute. Three test specimens per mix composition were tested for unconfined compressive strength at the end of the moist curing periods of 7 and 28 days.

#### 2.3.2. Linear Expansion

Linear expansion of the test specimens was determined in accordance with BS EN 13286-49 [30] by employing Perspex cells as schematically shown in Figure 1. Measurement of the expansion started after 7 days of moist curing, where approximately 10 mm of the top and bottom of two compacted cylindrical specimens per mix composition were unwrapped using a sharp razor to carefully remove the cling film. Thereafter, the test specimens were individually placed on a porous disc on top of a plastic platform in a Perspex cell. The Perspex cells were then covered with lids equipped with a dial gauge to measure the vertical expansion of the test specimens. The dial gauges were then adjusted to work appropriately by touching the top Perspex disc. Once the initial recording of the reading was completed, water was added to the Perspex cells through a top inlet, using a siphon to minimize any disturbance of the placed test specimens. The water level was then increased up to 10 mm of the test specimen bottom base. This process of partial immersion of specimen in water was monitored throughout the observing period.

## 3. Results

### 3.1. Unconfined Compressive Strength (UCS)

Figure 2 shows the results of the unconfined compressive strength of the various Kaolinite clay formulations at seven and 28 days of moist curing. Overall, all the kaolinite clay formulations experienced an increase in strength development at all the moist curing ages, with a higher gain in strength at 28 days of moist curing.

It can be observed that the highest seven- and 28-day UCS was achieved by the 1.2-40-60 mix. In contrast, the 1.4-40-60 mix experienced the lowest seven-day and 28-day UCS. It can also be noticed that the 1.2 OMC achieved significantly higher seven-day and 28-day UCS than the 1.4 OMC. The mellowed test specimens, generally, achieved a higher seven- and 28-day UCS than the control. It is also evident that mellowing using an initial moisture content of 40% and final moisture content of 60% after mellowing at 1.2 OMC achieved better performance than a 30% and 70% mellowing procedure at 1.4 OMC in terms of the seven- and 28-day UCS.

### 3.2. Swell Development of the Mellowed and Unmellowed Test Specimens

Figure 3 presents the typical linear expansion plots recorded throughout the soaking period for the different cylinder specimens produced with kaolinite clay dosed with 6 wt.% gypsum, stabilized with 2 wt.% lime and 2 wt.% SF at different moisture conditions with and without mellowing.

Overall, the 1.4 OMC performed better than the 1.2 OMC in terms of swelling. Mellowing using the moisture content of 40% and 60% achieved higher resistance to swelling than both mellowed specimens at 30% and 70% and the control specimens. It is evident that the mellowed test specimens 1.2-30-70 achieved the highest expansion throughout the observation period, while the 1.4-40-60 mellowed mixes experienced the lowest expansion magnitudes throughout the observation period. There was no evidence that the use of SF in the stabilization process exacerbated the linear expansion of the stabilized soil.

## 4. Discussion

Naturally, clay soils are rather malleable compacted mixtures of particles that are negatively charged [5]. Once clay soils are soaked in water, these particles attract water molecules, thereby altering the electrochemical interparticle of the equilibrium forces and resulting in an inter-crystalline expansion [31]. In the case of quicklime being introduced to clay soil, lime hydration would take place first, whose slaking results in heat that would dry out the soil clay particles. Once this reaction is completed, the hydrated lime formed dissolves and releases calcium ions [32]. In non-sulphate clay soils, these calcium ions get attached to the surface of the soil particles, thereby replacing most of the exchangeable ions [33]. The replacement of ions balances the electrostatic charges of the soil, resulting in the flocculation and agglomeration of soil particles [33]. In addition to the release of calcium ions, hydroxides are also introduced to the equation, leading to an increased pH value of up to 12.4 [34]. In the case of the inclusion of pozzolan binders, such as SF, GGBS, and PFA, in the stabilization process, the increase in the pH facilitates the dissolution of silicate (Si) and aluminate (Al) ions. This process would be followed by a pozzolanic reaction between the dissolved Al and Si ions and the excess calcium ions. Different hydrated products are then produced due to the pozzolanic reactions, such as calcium silicate hydrate (CSH), calcium aluminate hydrate (CAH), and calcium alumino-silicate hydrate (CASH) [10].

The presence of calcium hydroxide (portlandite) in clay soil stabilized with lime only prevents the dissolution of kaolin minerals [35], leading to reduced cohesion of the clay system [36], thereby inducing a compromise in strength. Nevertheless, the addition of pozzolan materials such as SF contributes to a higher degree of strength, leading to the acceleration of calcium hydroxide consumption, hence facilitating the formation of further hydrates, such as C-S-H and C-A-H. These additional hydrated products block the pores, reduce the permeability, enhance the porosity, and densify the clay soil system, thereby aiding the stability of the soil clay system under water-soaking conditions [37]. These developments significantly improved strength and swelling.

The change in volume (swelling) in sulphate-bearing clay is mainly due to sulphate modifying the cementitious products produced as a result of the reaction between the stabilizing agent and the clay soil. This reaction between sulphate, calcium and alumina in the presence of water forms expansive compounds such as ettringite [Ca_6_Al_2_(SO_4_)_3_(OH)_12_·26H_2_O] nucleates [38]. Wild et al. [39] reported the formation of a complex product of calcium-sulpho-alumino-silicate hydrate (C-A-S-S-H) on the surface of the clay plates of Lower Oxford Clay Soils. A continuous process of soil modification complemented by the pozzolanic activity occurred from the crystalline reactions (C-A-S-S-H) and the excess lime [1]. Only a limited amount of ettringite crystals is formed at the earlier moist curing age of 7-days. This is because the amount of soluble sulphate is limited due to the rather low solubility of gypsum along with the insufficient moisture content needed for the complete dissolution of gypsum. Ettringite crystals are very small needle-like and colloidal crystals with a long columnar morphology [39]. The nucleation and growth of ettringite crystals is beneficial as it improves the clay soil strength through the reduced porosity of the whole matrix by the interlocking behavior of the crystals formed [39].

Mellowing is thought to have further modification to the engineering properties of the stabilized sulphate clay soil. The ettringite crystals formed during the mellowing process allow expansive reactions prior to compaction. This leads to the consumption of both calcium and sulphate in the process, resulting in slightly reduced pH levels in comparison with unmellowed specimens [40]. The test results showing that the mellowed specimens exhibited better resistance to swelling and higher strength (in the case of 1.2 OMC) suggest that the chemical reactions occurring during the mellowing period are quite critical and an influencing factor [36,41,42,43]. The calcium ions are consumed firstly during the flocculation of the mixed materials during mellowing and secondly in the formation of ettringite. In sulphate clay soils, sulphate will also be consumed in the mellowing process of ettringite formation, leading to a relatively lower availability of sulphate ions in the mellowed specimens once soaked in water [41,42,43]. Thus, the reduced expansion encountered in the mellowed specimens might be attributed to the depletion of calcium and sulphate during mellowing because of the formation of the crystalline ettringite during the mellowing process [40]. Consequently, the amount of the ettringite crystals formed is reduced upon soaking in water, thereby reducing expansion. Another contributing factor could be that mellowing allows the formation of unhindered ettringite in a less restricted (uncompacted) environment, thus leading to less expansion and disruption in the mellowed specimens after compaction and soaking in water [41].

The establishment of OMC in complex mixtures in soil stabilization is one of the main technical problems researchers face when the soil stabilization process in road construction involves the inclusion of by-product additives and/or industrial waste products.

In soil stabilization, generally, the extent to which air voids can be removed depends mainly on the friction between clay soil particles and/or the clay soil strength [40]. These two factors are primarily governed by the moisture content of the clay soil during its compaction. The compaction level of clay soil is mainly affected by its moisture content. For instance, if the clay soil is in a completely dried state, it will not be compacted to its ultimate possible state of density [6]. However, once the moisture content increases, the soil gets lubricated. Hence, it will be easily compacted with its density increased. Whereas, at low water content, the attraction forces are large in the absorbed water layer, and thus there is higher resistance to the movement of the particles [18].

The results suggest that the use of 1.4 OMC induced a compromise on the UCS of all the mellowed and unmellowed specimens. This might be attributed to the higher moisture content that lowered the inter-particle friction and resulted in relatively poorer particle packing and interlocking [18]. This led to an enlargement in voids, which in turn influenced the matric suction and increased the system porosity, thereby lowering the resistance of the specimen to loading [44,45]. Another contributing factor to the reduced UCS of the 1.4 OMC specimens is the growth and development of the ettringite crystals instantly after blending sulphate soil with lime [39]. The higher moisture content induced resistance to compaction, thereby resulting in lowered density and reduced UCS. Gu et al. [40] stated that using additional moisture content above the OMC facilitates the formation of expansive brucite and hydrotalcite-like phases through delayed hydration, which leads to a reduction in UCS through crack formations.

The observed reduction in expansion in the case of the 1.4 OMC test specimens compared to the 1.2 OMC specimens is thought to be due to the increase in the inter-particle void spaces [18]. This could also be further explained as, at higher moisture content, the maximum densification degree could not be achieved. Thus, further space is formed to accommodate the ettringite crystals, which ultimately leads to lesser overall swelling magnitude [6].

## 5. Conclusions

In practice, the moisture content during compaction of unsterilized soils does not exceed 1.2 OMC due to concerns regarding strength reduction. In road design and construction, other measures can be taken to mitigate against swelling, such as sloping the roads appropriately and/or use of road surfacing. This appears to suggest a similar approach for stabilized and mellowed soils, including when SF is used. The present study examined the effects of mellowing and moisture content on the swelling and unconfined compressive strength performance of stabilized sulphate-bearing clay soil that was artificially dosed with 6 wt.% gypsum. Lime-SF blended stabilizer was used for stabilization. The study suggested two major outcomes:Overall, a trade-off between performance based on strength development and swelling potential was made. It was found more practical to aim for high strength and then to explore methods to mitigate against linear expansion.The findings of this study showed a significant drop in compressive strength when the targeted materials were mellowed at 1.4 OMC. Therefore, it is recommended to compact the materials at lower moisture condition, such as 1.2 OMC (as opposed to say 1.4 OMC), but mellow at the higher initial moisture condition of approximately 40% OMC (as opposed to say at 30% OMC).

## Figures and Tables

**Figure 1 materials-16-02187-f001:**
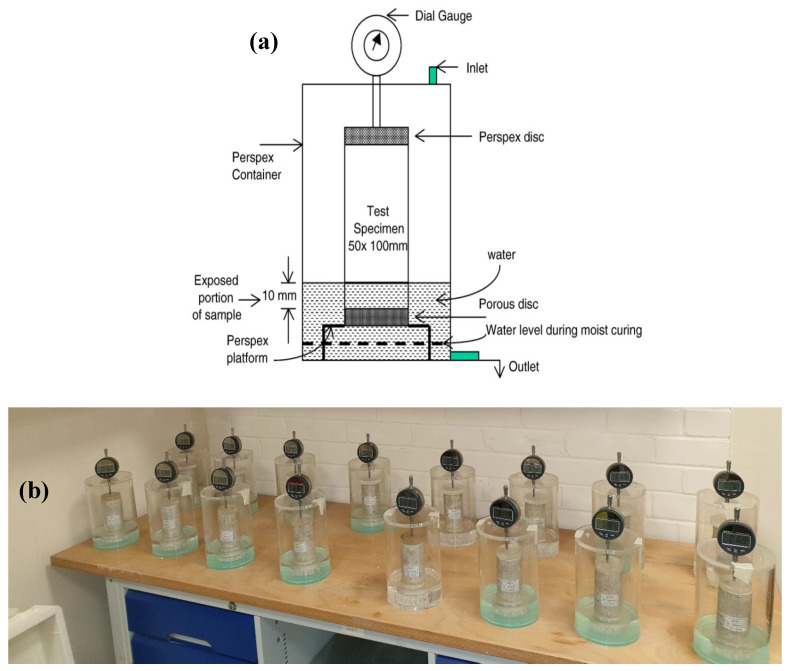
Showing (**a**) schematic diagram of a Perspex cell and, (**b**) group test set-up used for determining of linear expansion.

**Figure 2 materials-16-02187-f002:**
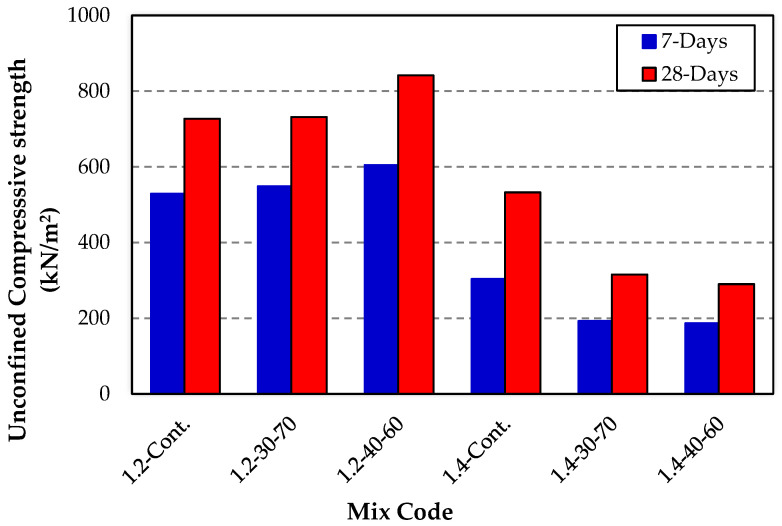
Unconfined 7- and 28-day compressive strength for various kaolinite clay cylinder specimens dosed with 6 wt.% gypsum and at varying OMCs.

**Figure 3 materials-16-02187-f003:**
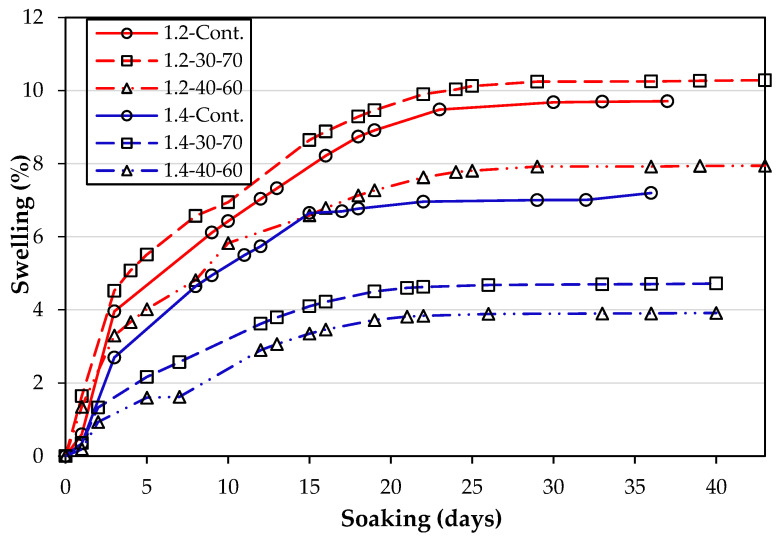
Linear expansion plots for Kaolinite clay specimens dosed with 6 wt.% Gypsum, at varying OMCs.

**Table 1 materials-16-02187-t001:** The different oxide compositions of some of the raw materials used in this study.

Oxides	Composition (%)		
	Kaolin	Lime	Silica Fume
CaO	<0.01	71.56	0.2
MgO	0.21	0.58	0.1
Al_2_O_3_	47.32	0.67	98.4
SiO_2_	35.96	0.07	0.2
Na_2_O	0.07	<0.02	-
P_2_O_5_	0.12	0.03	0.03
Fe_2_O_3_	0.69	0.05	0.01
Mn_2_O_3_	0.02	0.02	-
K_2_O	1.8	<0.01	0.2
TiO_2_	0.02	<0.01	-
V_2_O_5_	<0.01	0.02	-
BaO	0.07	<0.01	-
SO_3_	0.01	0.19	0.1
SO_4_	0.1	-	-
S	0.1	-	-

**Table 2 materials-16-02187-t002:** Grading and some other physical properties of the raw materials.

Properties	Kaolin	Lime	Silica Fume
**Particle size distribution**			
D_60_ (mm)	0.012	0.35	0.1
D_50_ (mm)	0.0085	0.25	0.04
D_30_ (mm)	0.0062	0.06	0.025
D_10_ (mm)	0.0018	<0.01	0.01
**Other properties**			
Linear shrinkage (%)	10.8	-	-
Linear expansion (%)	6.2	-	-
Swelling pressure (kPa)	1.3	-	-
Bulk density (kg/m^3^)	-	480	300
Specific gravity (mg/m^3^)	2.14	2.82	3.15
Alkalinity value (pH)	5.37	12.62	7
Colour	White	White	Grey
Loss on ignition (LOI)	13.1	27.4	0.5

**Table 3 materials-16-02187-t003:** Mix compositions of the various sulphate induced kaolinite clay mixes at varying OMC (1.2 and 1.4), stabilised with a blended binder of 2 wt.%lime-2 wt.%SF.

Mix Code	Elaborated Abbreviation	Mix Ingredients (g) per Specimen				Moisture Content (MC)		
		Kaolin	Gypsum	Lime	SF	Total MC	MC Pre-Mellowing	MC After-Mellowing
**1.2-Cont.**	K6G-2L2S-1.2OMC-Control	283	18.9	6.3	6.3	100.6	100.6	100.6
**1.2-30-70**	K6G-2L2S-1.2OMC (30%70%M)	283	18.9	6.3	6.3	100.6	30.2	70.4
**1.2-40-60**	K6G-2L2S-1.2OMC (40%60%M)	283	18.9	6.3	6.3	100.6	40.2	60.4
**1.4-Cont.**	K6G-2L2S-1.4OMC-Control	270.7	18	6	6	114.3	114.3	114.3
**1.4-30-70**	K6G-2L2S-1.4OMC (30%70%M)	270.7	18	6	6	114.3	34.3	80
**1.4-40-60**	K6G-2L2S-1.4OMC (40%60%M)	270.7	18	6	6	114.3	45.7	68.6

OMC—Optimum moister content, M—mellowing, K—kaolin, G—gypsum, L—lime, SF—silica fume.

## Data Availability

The data presented in this study are not publicly available due to ongoing research in this field.

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
