# Peer review of "Effects of Mellowing Practice on the Strength and Swelling Properties of Road Construction Materials: Case of Sulphate-Bearing Clay Soils Stabilised with Lime-Silica Fume Blended Binder"

_materials, 2023, doi:10.3390/ma16062187_

Round 1
Reviewer 1 Report
Please see the attachment for comments.

Author Response
The authors would like to appreciate the kind suggestions/recommendations from the reviewer, which have been addressed one after the other in the revised version of the proposed article. The changes made are marked up for clarity in the revised draft article.
Reviewer 2 Report
This paper aims at investigating the benefits of adopting mellowing, and also the optimal moisture condition for mellowing kaolinite clay. However, there are some obvious mistakes in this manuscript. The authors need to correct these errors and pay attention to the format problem.
1. Line 28, page 1: Please delete the dashed line in "com-pacting".
2. Table 3, page 4: The mix code "1.2-40-70" is incorrect and needs to be changed to "1.2-40-60".
3. The authors need to unify the style of the table. The title and borders of some tables are not aligned.
4. Line 211-213, page 6: Mellowing using an initial moisture content of 40% and 60% after mellowing did not achieve better performance than a 30% and 70% mellowing procedure at 1.4 OMC. Please provide an explanation.
5. Line 226, page 7: "both mellowed specimens at 30% and 60% and the control specimens", Please change "60%" to "70%".
6. Please compare the influence of remainder of 70 and 60 wt.% of the OMC on kaolinite clay after mellowing.

Author Response

(The authors gave the same response as above.)

Reviewer 3 Report
Interesting article, contains key research, although others would be useful too. Be sure to supplement the article with photos of samples, a test stand.
The aim of the research is to analyze swelling and moisture content in clay soils.The topic is interesting, it belongs to a very specific scientific field. Additional tests for clay soils are shown. Complete the article with photos of samples, apparatus and a description of equipment and sample preparation.
Although more publications by authors from Europe can be shown. Complete the article with photos of samples and apparatus.
Author Response
The authors would like to appreciate the kind suggestions/recommendations from the reviewer, which have been addressed one after the other in the revised version of the proposed article. The changes made are marked up for clarity in the revised draft article.
Furthermore, the authors have included a photo showing the sample with apparatus in the revised paper. Also, a detailed description of the sample preparation and engineering tests with their respective equipment had been provided in section 2.2 and 2.3.